# Learning Individually Inferred Communication for Multi-Agent Cooperation

**Ziluo Ding**    **Tiejun Huang**    **Zongqing Lu**[†]
Peking University
{ziluo,tjhuang,zongqing.lu}@pku.edu.cn

## Abstract

Communication lays the foundation for human cooperation. It is also crucial for multi-agent cooperation. However, existing work focuses on broadcast communication, which is not only impractical but also leads to information redundancy that could even impair the learning process. To tackle these difficulties, we propose *Individually Inferred Communication* (I2C), a simple yet effective model to enable agents to learn a prior for agent-agent communication. The prior knowledge is learned via causal inference and realized by a feed-forward neural network that maps the agent's local observation to a belief about who to communicate with. The influence of one agent on another is inferred via the joint action-value function in multi-agent reinforcement learning and quantified to label the necessity of agent-agent communication. Furthermore, the agent policy is regularized to better exploit communicated messages. Empirically, we show that I2C can not only reduce communication overhead but also improve the performance in a variety of multi-agent cooperative scenarios, comparing to existing methods.

## 1  Introduction

Deep reinforcement learning has achieved remarkable success in a series of challenging tasks, *e.g.*, game playing [12, 16, 23]. However, many real-world applications require multiple agents to learn to solve tasks collaboratively, such as autonomous driving [15], smart grid control [26], and intelligent traffic control [25]. Unfortunately, there exist several challenges impeding the breakthroughs in multi-agent reinforcement learning (MARL). On one hand, during training, the agent policy keeps updating, leading to non-stationary environment and unstable model convergence. On the other hand, even for the centralized training and decentralized execution (CTDE) paradigm which is designed to mitigate non-stationarity, such as MADDPG [10], COMA [4], VDN [20], QMIX [14], QTRAN [18], and MAAC [5], it is still hard for agents to act cooperatively during execution, since partial observability and stochasticity can easily break the learned cooperative strategy and result in catastrophic miscoordination [24].

Communication lays the foundation for human cooperation [2]. It also holds for multi-agent cooperation. Communication could help agents form a good knowledge of cooperative strategies. Recently, many researches focus on trainable communication channel which agents can use to obtain extra information during both training and execution to tackle the challenges aforementioned. For most existing work, such as [19, 17, 1, 27], once the decision of communication is made, messages will be broadcast to all other/predefined agents. However, this requires lots of bandwidth and incurs additional delay in practice. More importantly, not every agent can provide useful information and redundant information could even impair the learning process [21, 8]. These limitations make it a less than ideal communication paradigm. Furthermore, it has never been the way how humans

---

[†]Corresponding author

communicate, since humans would use their prior knowledge to choose whom to communicate with if necessary. For example, one of the most important features in Quora, a question-and-answer website, is *ask to answer*. One can invite the person who is believed to be highly relevant to answer the question. Otherwise, one could be overwhelmed by massive answers, not to mention most are useless or misleading. Therefore, we argue that an excellent communication paradigm should take into consideration finding the right ones to communicate by enabling the agent to figure out who are highly relevant with its situation and ignore those are not.

In this paper, we propose *Individually Inferred Communication* (I2C), a simple yet effective model to enable agents to learn a prior for agent-agent communication. More specifically, each agent is capable of exploiting its learned prior knowledge to figure out which agent is *relevant and influential* by just local observation. The prior knowledge is learned via causal inference and realized by a feed-forward neural network that maps its local observation to a belief about who to communicate with. The influence of one agent on the other is captured by the causal effect of the agent's action on the other's policy. For any agent that can cause drastic change to the other's policy, that agent is considered as relevant and influential. Unlike existing work [6] that utilizes social influence for reward shaping and hence encourages the agent to influence the behaviors of others, we quantify the causal effect inferred via the joint action-value function to label the necessity of agent-agent communication and thus a prior can be learned by supervised learning. Furthermore, correlation regularization is proposed to help the agent learn a better policy under the influence of the agents it communicates with.

I2C learns one-to-one communication instead of one/all-to-all communication adopted by existing work [19, 17, 1, 27], and hence makes agents focus only on relevant information. In addition, I2C restricts the communication range of agent to the field of view. These together make I2C much more practical, since in real-world applications communication is always limited by bandwidth and range. As I2C infers the causal effect between agents by only the joint action-value function, it is compatible with many CTDE frameworks. Moreover, by alternatively capturing the casual effect of communicated messages, I2C can also serve as communication control to reduce overhead for full communication methods, *e.g.*, TarMAC [1]. We implement I2C based on two CTDE frameworks to learn communication in MARL and evaluate it in three classic multi-agent cooperative scenarios, *i.e.*, cooperative navigation, predator prey, and traffic junction. In cooperative navigation and predator prey, we empirically demonstrate that I2C outperforms the existing methods with/without communication. By ablation studies, we confirm that the inferred causal effect indeed captures the necessity of agent-agent communication, cooperative strategy can be better learned by agent-agent communication, and correlation regularization indeed helps the agent develop better policy. In traffic junction, I2C is implemented as communication control for full communication methods, and it is shown that I2C can not only reduce communication overhead but also improve the performance.

To the best of our knowledge, I2C is the *first* work that learns one-to-one communication in MARL.

## 2 Related work

Learning communication in MARL has been greatly advanced recently. DIAL [3] is proposed as a simple differential communication module which allows the gradient to flow between agents for training, but it is limited to discrete message. CommNet [19] is a large connected structure that controls all the agents. It averages the hidden layers from all the agents to produce the message, which however inevitably leads to information loss. BiCNet [13] trains a bi-directional communication channel using recurrent neural networks to integrate messages from all the agents. Both CommNet and BiCNet use a communication channel to connect all the agents, *i.e.*, each agent sends message to and receives from all other agents. In such a case, there is no doubt that agents can be flooded by information as the number of agents grows.

IC3Net [17], an upgraded version of CommNet, tries to instruct agents to learn when to communicate by bringing in a single gating mechanism. IC3Net makes progress on considering that communication is not always necessary, where each agent determines whether to send messages to the others at each timestep, but what if an agent's message is instructive to some agents but harmful to others. SchedNet [9], on the other hand, is a weight-based scheduler to decide which agents should be entitled to broadcast their messages. SchedNet needs the observations from all the agents as input, making it highly unrealistic. ATOC [8] is a communication model which can be seen as a more complex gate mechanism to decide when to speak to predefined neighboring agents. VBC [27] extracts informative

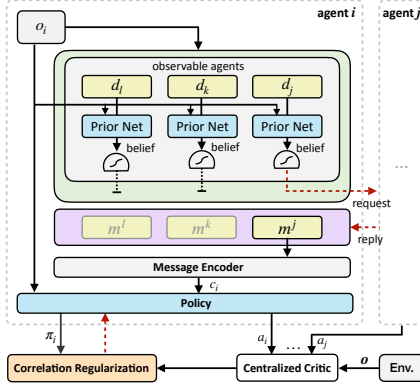

Figure 1: I2C's architecture.

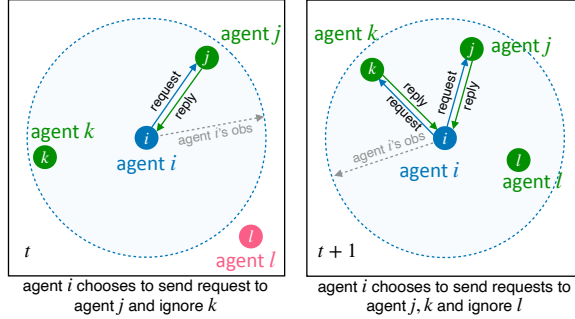

Figure 2: I2C's request-reply communication mechanism. The agent can only communicate with other agents in its field of view.

messages based on variance in action values, where each agent determines whether to broadcast to all other agents, then other agents decide whether to reply. However, VBC works only on the methods that factorize the joint action-value function, such as QMIX [14].

Inspired by the wide application of attention mechanism in computer vision and natural language processing [11, 22], many researchers raise a lot of interests in its ability of discerning what part of input should be paid more attention. DGN [7] harvests information from messages in the neighborhood by convolution with multi-head attention kernels. However, the communication of DGN is limited by the number of convolutional layers. TarMAC [1] pilots agents to form targeted communication behavior also through multi-head attention. Basically, TarMAC is still a traditional broadcast approach (all-to-all communication) with attention that allows agents to turn a blind eye to received inconsequential messages.

None of existing work completely abandons broadcast mechanism. However, I2C enables the agent to make a judgment only based on its own observation, then it communicates only with the relevant and influential agents within its field of view. We will show that such a communication mechanism is actually able to obtain useful information and discard redundant and useless information meanwhile.

## 3 Methods

I2C can be instantiated by any framework of CTDE with a joint action-value function. I2C consists of a prior network, a message encoder, a policy network, and a centralized critic, as illustrated in Figure 1. We consider fully cooperative multi-agent tasks that are modeled as Dec-POMDPs augmented with communication. Agents together aim to maximize the cumulative team reward. Additionally, agents are able to communicate with observed agents and adopt the request-reply communication mechanism, which is illustrated in Figure 2. At a timestep, each agent $i$ obtains a partial observation $o_i$ and identifies which agents are in the field of view based on $o_i$. Suppose agent $j$ can be seen by agent $i$, the prior network $b_i(\cdot)$ takes as input local observation $o_i$ and ID $d_j$ (or any feature that identifies agent $j$) and outputs a belief indicating whether to communicate with agent $j$. Based on the belief, if agent $i$ sends a request (scalar) to agent $j$ via a communication channel, agent $j$ will respond with a message $m^j$, *i.e.*, its own (encoding of) observation. All the messages received by agent $i$, $\boldsymbol{m}_i$, are fed into the message encoder network $e_i(\cdot)$ to produce the encoded message $c_i$, and the policy network outputs the distribution over actions $\pi_i(a_i|c_i, o_i)$. The centralized critic approximates the joint action-value function.

### 3.1 Learning Prior Network via Causal Inference

The key component of I2C is the prior network, which enables agents to have a belief about who to communicate with. Intuitively, an agent is more likely to communicate with those who are potentially imposing more influence on its strategy, hoping to get clues about how they tend to act and how to react cooperatively. Therefore, the causal effect of other agent can be regarded as the necessity of conditioning on other agent's action for decision making. Fortunately, we can measure and quantify the causal effect between agents via the centralized critic and train the prior network to determine agent-agent communication.

Assume there are two conditional probability distributions over the action space of agent $i$, *i.e.*, $P(a_i|\boldsymbol{a}_{-i}, \boldsymbol{o})$ and $P(a_i|\boldsymbol{a}_{-ij}, \boldsymbol{o})$, where $\boldsymbol{o}$ denotes the joint observations of all agents, $\boldsymbol{a}_{-i}$ denotes the given joint actions of all agents except for agent $i$, and similarly $\boldsymbol{a}_{-ij}$ denotes the given joint actions except for agent $i$ and $j$. The latter probability distribution does not condition on agent $j$'s action compared with the former one, implying agent $i$ ignores agent $j$ when making decision. Then, the causal effect $\mathcal{I}_i^j$ of agent $j$ on agent $i$ can be defined as:

$$\mathcal{I}_i^j = D_{\mathrm{KL}}\left(P(a_i|\boldsymbol{a}_{-i}, \boldsymbol{o})\|P(a_i|\boldsymbol{a}_{-ij}, \boldsymbol{o})\right).$$

Kullback-Leibler (KL) divergence is employed to measure the discrepancy between these two conditional probability distributions. Note that the conditioned actions of other agents $\boldsymbol{a}_{-i}$ are the actions sampled from their current policies. The magnitude of $\mathcal{I}_i^j$ indicates how much agent $i$ will make adjustments to its policy if it takes into account the action of agent $j$, and also shows how much the strategy of agent $j$ is correlated with the policy of agent $i$.

The joint action-value function is exploited to calculate the two probability distributions. $P(a_i|\boldsymbol{a}_{-i}, \boldsymbol{o})$ is calculated as the softmax distribution over the actions of agent $i$,

$$P(a_i|\boldsymbol{a}_{-i}, \boldsymbol{o}) = \frac{\exp(\lambda Q(a_i, \boldsymbol{a}_{-i}, \boldsymbol{o}))}{\sum_{a_i'} \exp(\lambda Q(a_i', \boldsymbol{a}_{-i}, \boldsymbol{o}))},$$

where $Q(\boldsymbol{a}, \boldsymbol{o})$ is the joint action-value function and $\lambda \in \mathbb{R}^+$ is the temperature parameter. $P(a_i|\boldsymbol{a}_{-ij}, \boldsymbol{o})$ can be regarded as the marginal distribution of $P(a_i, a_j|\boldsymbol{a}_{-ij}, \boldsymbol{o})$ and computed as,

$$P(a_i|\boldsymbol{a}_{-ij}, \boldsymbol{o}) = \sum_{a_j} P(a_i, a_j|\boldsymbol{a}_{-ij}, \boldsymbol{o}) = \sum_{a_j} \frac{\exp(\lambda Q(a_i, a_j, \boldsymbol{a}_{-ij}, \boldsymbol{o}))}{\sum_{a_i', a_j'} \exp(\lambda Q(a_i', a_j', \boldsymbol{a}_{-ij}, \boldsymbol{o}))}.$$

Then, we can obtain the causal effect $\mathcal{I}_i^j$ of agent $j$ on $i$ under current state. As agent $i$ only has $o_i$ to determine agent-agent communication, we store $\{(o_i, d_j), \mathcal{I}_i^j\}$ as a sample of training data set $\mathcal{S}$ for the prior network. The training will be discussed later.

**Communication Reduction.** The prior network is primarily designed to initiate agent-agent communication. However, it can alternatively serve as a component to reduce communication for full communication methods, *e.g.*, TarMAC [1], where there is a joint action-value function that additionally takes received messages of all agents as input, $Q(\boldsymbol{a}, \boldsymbol{o}, \boldsymbol{m})$. With this joint action-value function, we can also measure the casual effect of communicated messages. More specifically, two conditional probability distributions can be calculated: $P(a_i|\boldsymbol{a}_{-i}, \boldsymbol{o})$ and $P(a_i|\boldsymbol{a}_{-i}, \boldsymbol{m}_i, \boldsymbol{o})$. The KL divergence between them measures the causal effect of received messages $\boldsymbol{m}_i$ on agent $i$, which similarly can be employed to determine the necessity of existing communication. The effect of I2C in reducing communication is also investigated in the experiments.

### 3.2 Correlation Regularization

The causal effect is inferred via the joint action-value function to determine the necessity of communication between agents. Then, each agent takes action based on its observation with/without communicated messages in a decentralized way. This can be seen as employing decentralized policies augmented by communication to approximate the centralized policy derived from the joint action-value function. With such an approximation, ideally the policy of an agent requesting communication should condition on the observation and action of the communicated agent. However, it is impossible to send action directly in practice, otherwise circular dependencies can occur. Therefore, the agent policy has to condition only on the observation. Nevertheless, we design correlation regularization to help the agent correlate other agent's observation and action and thus correct the discrepancy between the policies with/without considering the action.

As illustrated in Figure 2 (*right*), agent $i$ perceives agent $j$, $k$, $l$ and chooses to send requests to agent $j$, $k$ based on the prior knowledge. Then, agent $i$ takes action based on the observations of agent $i$, $j$ and $k$. To encourage the agent to learn inferring others' actions from their observations, we force the policy conditioned on the observations $\pi_i(a_i|e_i(o_j, o_k), o_i)$ be close to the policy also conditioned on the action $\hat{\pi}_i(a_i|a_j, a_k, e_i(o_j, o_k), o_i)$. From the perspective of agent $i$, $\hat{\pi}_i(a_i|a_j, a_k, e_i(o_j, o_k), o_i)$ is exploited to approximate $P(a_i|\boldsymbol{a}_{-i}, \boldsymbol{o})$, and thus we directly use $P(a_i|\boldsymbol{a}_{-i}, \boldsymbol{o})$ as the target of $\pi_i(a_i|e_i(o_j, o_k), o_i)$. The KL divergence between these two distributions, $D_{\mathrm{KL}}(P(a_i|\boldsymbol{a}_{-i}, \boldsymbol{o})\|\pi_i(a_i|e_i(o_j, o_k), o_i))$ is employed as the regularization to the policy.

## 3.3 Training

The centralized joint action-value function $Q^{\boldsymbol{\pi}}(\boldsymbol{a}, \boldsymbol{o})$, which is parameterized by $\theta_Q$ and takes as input the actions and observations of all the agents, guides the policy optimization. The centralized critic is updated as

$$\mathcal{L}(\theta_Q) = \mathbb{E}_{\boldsymbol{o},\boldsymbol{a},r,\boldsymbol{o}'}[(Q^{\boldsymbol{\pi}}(\boldsymbol{a}, \boldsymbol{o}) - y)^2],$$
$$y = r + \gamma Q^{\boldsymbol{\pi}}(\boldsymbol{a}', \boldsymbol{o}')|_{\boldsymbol{a}' \sim \boldsymbol{\pi}(\boldsymbol{o}')},$$

where $\boldsymbol{a}'$ are sampled from $\boldsymbol{\pi}(\boldsymbol{o}')$. The regularized gradient of each policy network parameterized by $\theta_{\pi_i}$ can be written as,

$$\nabla_{\theta_{\pi_i}} \mathcal{J}(\theta_{\pi_i}) = \mathbb{E}_{\boldsymbol{o},\boldsymbol{a}}[\mathbb{E}_{\pi_i}[\nabla_{\theta_{\pi_i}} \log \pi_i(a_i|c_i, o_i) Q^{\boldsymbol{\pi}}(a_i, \boldsymbol{a}_{-i}, \boldsymbol{o})] - \eta \nabla_{\theta_{\pi_i}} D_{\mathrm{KL}}(\pi_i(\cdot|c_i, o_i) \| P(\cdot|\boldsymbol{a}_{-i}, \boldsymbol{o}))],$$

where $\eta$ is the coefficient for correction regularization. By the chain rule, the gradient of each message encoder network parameterized by $\theta_{e_i}$ can be further derived as,

$$\nabla_{\theta_{e_i}} \mathcal{J}(\theta_{e_i}) = \mathbb{E}_{\boldsymbol{o},\boldsymbol{m},\boldsymbol{a}}[\mathbb{E}_{\pi_i}[\nabla_{\theta_{e_i}} e_i(c_i|\boldsymbol{m}_i) \nabla_{c_i} \log \pi_i(a_i|c_i, o_i) Q^{\boldsymbol{\pi}}(a_i, \boldsymbol{a}_{-i}, \boldsymbol{o})]$$
$$- \eta \nabla_{\theta_{e_i}} e_i(c_i|\boldsymbol{m}_i) \nabla_{c_i} D_{\mathrm{KL}}(\pi_i(\cdot|c_i, o_i) \| P(\cdot|\boldsymbol{a}_{-i}, \boldsymbol{o}))].$$

The prior network parameterized by $\theta_{b_i}$ is trained as a binary classifier using the training data set $\mathcal{S}$, and it updates with the loss,

$$\mathcal{L}(\theta_{b_i}) = \mathbb{E}_{(o_i, d_j), \mathcal{I}_i^j \sim \mathcal{S}}[-(1 - y_i^j) \log(1 - b_i(o_i, d_j)) + y_i^j \log(b_i(o_i, d_j))],$$

where $y_i^j = 1$ if $\mathcal{I}_i^j \geq \delta$, zero otherwise, and $\delta$ is a hyperparameter. The prior network can be learned end-to-end or in a two-phase manner. As for two-phase manner, phase one is to train the prior network with data generated from any pre-trained CTDE algorithm, and phase two is to train the rest of the architecture from scratch, with the prior network fixed. We found that the two-phase manner learns faster and thus is employed in the experiments.

## 4 Experiments

We evaluate I2C in three multi-agent cooperative tasks: *cooperative navigation*, *predator prey*, and *traffic junction*. For cooperative navigation and predator prey, I2C is built on MADDPG [10] to learn agent-agent communication. For traffic junction [19], our main purpose is to investigate the effectiveness of I2C on communication reduction, and thus we built I2C directly on TarMAC [1] and it serves as communication control. In the experiments, I2C and baselines are *parameter-sharing*. Moreover, to ensure the comparison is fair, their basic hyperparameters are the same and their sizes of network parameters are also similar. Please refer to the supplementary for the hyperparameter settings.

### 4.1 Cooperative Navigation

**Task and Setting.** In this task, $N$ agents try to occupy $L$ landmarks. Each agent obtains partial observation of the environment and it is only allowed to communicate with observed agents. The team reward is based on the proximity of agents to landmarks, which is the sum of negative distances of all landmarks to their closest agents. Moreover, agents are penalized for collision. In this task, each agent needs to consider the intentions of other agents to infer the right landmark to occupy and avoid collision meanwhile. In the experiment, we train I2C and baselines in the setting of $N = 7$ and $L = 7$ with random initial locations, and each agent observes relative positions of three nearest agents and three nearest landmarks. The collision penalty is set to $r_{\mathrm{collision}} = -1$. The length of each episode is 40 timesteps.

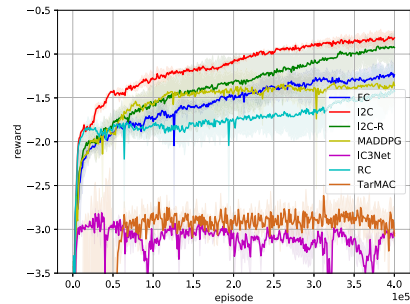

Figure 3: Reward of I2C against baselines during training in cooperative navigation. Shadowed area is enclosed by the min and max value of three training runs, and solid line in middle is mean.

**Quantitative Results.** We compare I2C with three existing methods: IC3Net [17], TarMAC [1] and MADDPG [10]. We do not choose [6] as baseline, because it focuses on

Table 1: Cooperative Navigation

|  | I2C | I2C-R | FC | RC | MADDPG | IC3NET | TARMAC |
|---|---|---|---|---|---|---|---|
| REWARD | $\mathbf{-0.73}$ $\pm0.20$ | $-0.77$ $\pm0.15$ | $-1.08$ $\pm0.22$ | $-1.30$ $\pm0.48$ | $-1.26$ $\pm0.26$ | $-3.03$ $\pm0.71$ | $-2.93$ $\pm0.26$ |
| OCCUPIED | $\mathbf{6.13}$ $\pm1.26$ | $5.85$ $\pm1.12$ | $4.69$ $\pm1.31$ | $4.79$ $\pm1.30$ | $4.25$ $\pm2.01$ | $0.76$ $\pm0.67$ | $0.88$ $\pm0.33$ |

social dilemma where each agent has an individual reward, not fully cooperative tasks we consider. Moreover, it assumes influence is unidirectional, meaning they set a disjoint set of influencer and influencee agents, and all influencers must act first. As I2C learns the prior network to determine whether to communicate with each observed agent, I2C is also compared against two baselines with only difference in communication: each agent always communicates with all observed agents, denoted as FC, and each agent randomly communicates with each observed agent with probability $p_{\mathrm{comm}}$, denoted as RC. Moreover, we also investigate I2C without correlation regularization, denoted as I2C-R. Figure 3 shows the learning curves of all the methods in terms of final reward in cooperative navigation. We can see that I2C converges to the highest reward compared with all other baselines. For testing, we evaluate all the methods by 100 test runs. Table 1 shows the results, including reward and occupied landmarks. I2C achieves the best performance in terms of both reward and occupied landmarks. TarMAC and IC3Net have the worst performance and they fail to develop cooperative strategy in this task. One of possible reasons is that they are poor at handling tasks where team reward cannot be decomposed to individual ones since TarMAC also struggles in all the scenarios of StarCraft II (cooperative game) [24]. As observed in the experiments, both TarMAC and IC3Net agents do not have a clear goal at the beginning so that they perform conservatively and aimlessly and prefer to avoid collisions rather than approach landmarks.

**Ablations.** As illustrated in Figure 3 and Table 1, FC achieves higher reward than MADDPG, demonstrating communication could help agents obtain valuable information and learn better policies. However, RC is worse than MADDPG, verifying that useless information can impair the performance. I2C outperforms FC. This demonstrates that there exists redundancy even in full communication with only observed agents and I2C is able to extract necessary communication for faster convergence and better performance. Moreover, in the experiments, RC is tuned by $p_{\mathrm{comm}}$ to have nearly the same amount of communication with I2C. However, I2C substantially outperforms RC. This verifies that the causal effect between agents truly captures the necessity of communication and the learned prior network is also effective. In addition, the superior performance of I2C to I2C-R proves the effectiveness of correlation regularization for learning better policy.

**Model Interpretation.** We first analyze the learned policies by I2C and the baselines. I2C agents have explicit targets to occupy and they tend to choose different landmarks as much as possible to ensure more landmarks can be occupied. However, the baselines do not develop such behaviors. Moreover, I2C also learns cooperative strategies. For example, one agent would abandon its target landmark and occupy other landmark while seeing there exists other agent having more advantages for occupying its target landmark, as illustrated in the top row of Figure 4. Moreover, one agent would give up the occupied landmark for other agent that possesses the same target so that they could together occupy the landmarks more quickly, as illustrated in the mid row of Figure 4. These cooperative strategies together form more sophisticated team strategies, as illustrated in the bottom

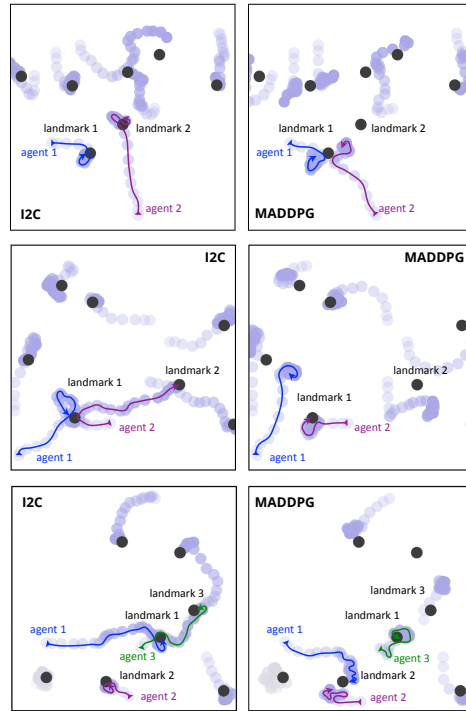

Figure 4: Illustration of learned cooperative behaviors of I2C agents comparing to MADDPG agents in cooperative navigation. Each row has the same initial positions of agents and landmarks, where black circles are landmarks and purple circles are trajectories of agents and darker circles are more recent agent positions.

row of Figure 4. The development of all these strategies requires inferring the intentions of other agents. However, without communication, it is hard to figure out other's intention. As illustrated in Figure 4, when two MADDPG agents approach a same target landmark, they always choose to share this landmark and none of them are willing to leave, regardless of collisions. On the other hand, IC3Net and TarMAC agents are very conservative. They behave hesitantly and wander around as soon as they find out other agent is targeting the same landmark to avoid collision. It seems the agent understands the intention of other agent, but it does not know how to cooperatively react.

We then investigate the communication pattern of I2C. Figure 5 illustrates the change of communication overhead as one episode progresses, where the communication overhead is the ratio between the sum of communicated agents and the sum of observed agents for all agents. As the episode starts, the communication overhead quickly increases to more than $80\%$. This is because at the beginning of an episode, agents need to determine targets and avoid target conflicts, and thus more communications are needed to reach a consensus. As the episode progresses, agents are moving more closely to their targets and agents become less influential on the policies of others. Thus, the communication overhead is also progressively reduced. Eventually, the communication overhead is reduced to the minimal, about only $10\%$.

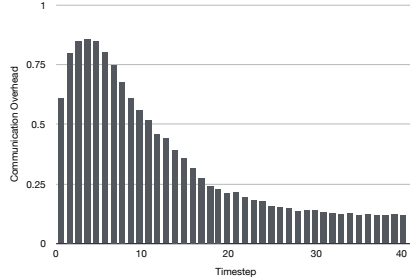

Figure 5: Change of communication overhead as one episode progresses in cooperative navigation.

## 4.2 Predator Prey

**Task and Setting.** In this task, $N$ predators (agents) try to capture $M$ preys. Each predator obtains partial observation and can only communicate with observed predators. Preys have a pre-defined area of activity and move in the opposite direction of the closest predator. Moreover, preys move faster than predators, and thus it is impossible for a predator to capture a prey alone. As for difference in predator-prey between ours and the IC3Net settings [17], ours has three moving preys, while IC3Net has only one prey and it is stationary. The team reward of predators is the sum of negative distances of all the preys to their closest predators. Predators are also penalized for colliding with other predator. In this task, predators need to learn how to cooperate with others to surround and capture preys. In the experiment, there are $N = 7$ predators and $M = 3$ preys with random initial locations, and each predator can observe relative positions of three nearest predators and three preys. The collision penalty is set to $r_{\text{collision}} = -1$, and one episode is 40 timesteps.

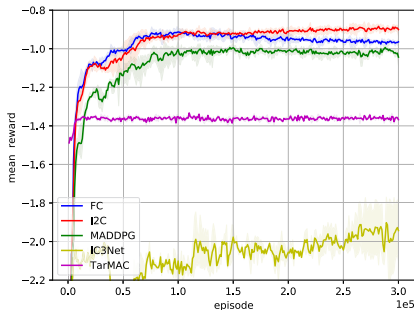

Figure 6: Mean reward of I2C against baselines during training in predator prey.

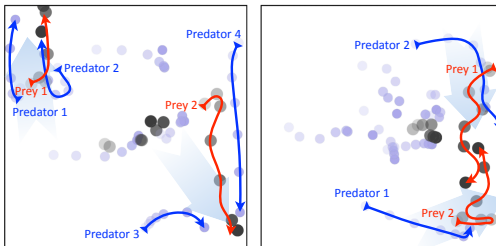

Figure 7: Illustration of learned strategies of I2C agents in predator prey. Black and purple circles are trajectories of preys and predators, respectively, and darker circles are more recent positions.

**Quantitative Results and Analysis.** We compare I2C with MADDPG, TarMAC, IC3Net and FC. As shown in Figure 6, I2C converges to the highest mean reward, averaged over timesteps. Table 2 shows the test results of 100 runs in terms of mean reward. I2C also performs best. Comparing to cooperative navigation, the difference between I2C and FC in both performance and communication overhead narrows down, where the communication overhead of I2C is about $48\%$. This is because communication is more crucial in predator prey and there is no much communication redundancy. More specifically, when a predator moves close to a prey, it is quite necessary to keep communicating

Table 2: Predator Prey

|  | I2C | FC | MADDPG | TARMAC | IC3NET |
|---|---|---|---|---|---|
| MEAN REWARD | $-\mathbf{0.83}\pm0.02$ | $-0.88\pm0.02$ | $-0.98\pm0.02$ | $-1.36\pm0.00$ | $-1.66\pm0.08$ |

with its close predators, because they need to cooperate with each other consistently to capture the highly active prey.

As observed in the experiments, I2C agents learn sophisticated cooperative strategies to capture preys. Normally, it is hard for less than three predators to surround a prey since there always exists a gap for the prey to escape. However, I2C agents learn to drive preys to the corner and exploit it to help them encircle preys, as illustrated in Figure 7 (*left*). Thus, even two I2C agents could capture a prey. In addition, I2C agents chasing different preys are prone to drive their preys together, so that a large encirclement can be formed, as illustrated in Figure 7 (*right*).

## 4.3 Traffic Junction

**Task and Setting.** In traffic junction [19], many cars move along two-way roads with one or more road junctions following the predefined routes. Cars only have two actions: brake (stay at its current location) and gas (move one step forward following the route). At each timestep, a new car enters the environment with probability $p_{\mathrm{arrive}}$ from each entry point until the total number of car reaches $N_{\mathrm{max}}$. After a car completes its route, it will be removed from the grid immediately, but it can be added back with a reassigned route. Cars will continue to move even after collisions. The agent gets a penalty $r_{\mathrm{collision}} = -10$ for colliding with other car and gets a reward of $-0.01\tau$ at every timestep, where $\tau$ is the number of timesteps since the car arrived. The team reward is the sum of all individual rewards.

We implement the *medium* and *hard* mode in the experiments. In the medium mode as illustrated in Figure 8a, there are one junction, four entry points, and three possible routes at each entry point, and $N_{\mathrm{max}} = 10$, $p_{\mathrm{arrive}} = 0.05$. In the hard mode as illustrated in Figure 8b, there are four junctions, eight entry points, and seven possible routes at each entry point, $N_{\mathrm{max}} = 20$ and $p_{\mathrm{arrive}} = 0.03$. In both modes, it turns out communication is trivial as long as cars have vision, since just one-grid vision provides cars enough reaction time to avoid collisions. Thus, all cars are set to only observe the grid it locates. In other words, cars only know the information of where they stay currently

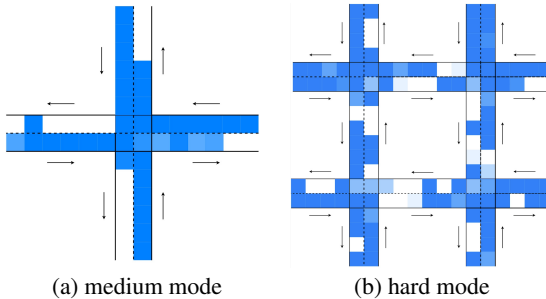

(a) medium mode      (b) hard mode

Figure 8: Communication overhead at different locations in the medium and hard mode of traffic junction. Darker color indicates higher communication overhead.

and are blind to anything else. In this setting, it is extremely hard for cars to finish the entire route safely without communication. Unlike previous two experiments, in traffic junction we measure the causal effect directly from messages so that the prior network predicts whether it is necessary to send requests to all other agents for communication at each timestep. In this setting, we investigate whether the causal effect inferred from messages can also help reduce communication for full communication methods. I2C is build on TarMAC and serves as communication control, denoted as I2C+TarMAC.

**Quantitative Results.** We compare I2C+TarMAC with TarMAC, IC3Net and no communication baseline in the medium and hard mode of traffic junction. Table 3 shows the performance of all the methods in terms of success rate of 10000 runs (an episode succeeds if it completes without any collisions) for both the medium and hard mode. In the medium mode, both I2C+TarMAC and TarMAC perform very well in success rate and reach above 97%. No communication baseline has a success rate of 79.19%, while IC3Net only gets 78.08%. In the hard mode, I2C+TarMAC is the only model that achieves more than 90% success rate, while TarMAC, IC3Net and no communication baseline have 89.24%, 40.47%, and 48.10%, respectively. This again verifies I2C can not only reduce communication but also improve performance, which is attributed to the prior network that accurately captures the necessity of communication. Note that the performance of TarMAC and IC3Net is different from that in their original paper, due to different experiment settings. We make heavier

Table 3: Traffic Junction

|  | I2C+TARMAC | TARMAC | IC3NET | NO COMMUNICATION |
|---|---|---|---|---|
| MEDIUM | **97.92%** | 97.60% | 78.08% | 79.19% |
| HARD | **92.17%** | 89.24% | 40.47% | 48.10% |

traffic and restrict cars' vision so as to demonstrate the superiority of I2C. They all have similar performance in easier settings.

**Model Interpretation.** Figure 8 shows the communication overhead at different locations for both modes. The communication overhead at a location is the ratio between communicating cars at the location and total cars that pass the location. In the medium mode, as illustrated in Figure 8a, the communication overhead is reduced by nearly $34\%$ compared to TarMAC. More specifically, cars communicate with other cars with high probability when moving towards and crossing the junction. However, communication occurs less after passing the junction. Intuitively, cars want to know as much information as possible, especially the location of other agents, in order to avoid collisions when crossing the junction. Even for the cars at the beginning of the route, they also need to prepare for the brake of front cars, which cannot be realized without communication. In contrast, the optimal action for all the cars is gas all the time after passing the junction. This strategy guarantees that no collisions occur and time delay is minimal, and thus communication is much less necessary.

Figure 8b shows the communication overhead for the hard mode, where the communication overhead is reduced by nearly $37\%$. Note that the hard mode is far more complicated than the medium mode (more junctions, entry points and routes), which makes communication more crucial for cars to avoid collision. Similarly, cars tend to communicate with other cars before entering all the four junctions. However, in the hard mode, keeping gas after crossing a junction may not be appropriate anymore since there may be another junction in the route. For the locations between two junctions, communication occurs with low probability. After crossing a junction, cars are prone to communicate less compared with other locations, which accords with the communication pattern shown in the medium mode.

## 5    Conclusions

We have proposed I2C to enable agents to learn a prior for agent-agent communication. The prior knowledge is learned via the causal effect between agents which accurately captures the necessity of communication. In addition, the agent policy is also regularized to better make use of communicated messages. Moreover, I2C can also serve as a component for communication reduction. As I2C relies on only a joint action-value function, it can be instantiated by many centralized training and decentralized execution frameworks. Empirically, it is demonstrated that I2C outperforms existing methods in a variety of cooperative multi-agent scenarios.

## Broader Impact

The experimental results are encouraging in the sense that we demonstrate I2C is a promising method for dealing with targeted communication in multi-agent communication based on causal influence. It is not yet at the application stage, and does not have broader impact. However, this work learns one-to-one communication instead of one/all-to-all communication, making I2C more practical in real-world applications.

## Acknowledgments and Disclosure of Funding

This work is supported by NSF China under grant 61872009.

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
