[Supplementary Material]

# A   Environmental Settings

**Cooperative Navigation and Predator Prey.** In cooperative navigation, there are 7 agents and the size of each is 0.05. They need to occupy 7 landmarks with size of 0.05. Each agent is only allowed to communicate with three closest agents. And the acceleration of agents is 0.7. In predator prey, the number of predators (agents) and preys is set to 7 and 3, respectively, and their sizes are 0.04 and 0.05. Each predator is only allowed to communicate with three closest predators. The acceleration is 0.5 for predators and 0.7 for preys. The team reward is similar for both tasks. At a timestep $t$, it can be written as $r_{\text{team}}^t = -\sum_{i=1}^{n} d_i^t + C^t r_{\text{collision}}$, where $d_i^t$ is the distance of landmark/prey $i$ to its nearest agent/predator, $C^t$ is the number of collisions (when the distance between two agents is less than the sum of their sizes) occurred at timestep $t$, and $r_{\text{collision}} = -1$. In addition, agents act discretely and have 5 actions (stay and move up, down, left, right).

**Traffic Junction.** Each car observes its previous action, route identifier, location, and a vector specifying sum of one-hot vectors for any car presented at that car's location. For medium mode, it has $14 \times 14$ grids consisting of one junction of two-way roads as shown in Figure 9 (*left*). The maximum number of steps is 40, $N_{\text{max}} = 10$ and $p_{\text{arrive}} = 0.05$. For hard mode, it has $18 \times 18$ grids consisting of four junctions of two-way roads. The maximum number of steps is 80, $N_{\text{max}} = 20$ and $p_{\text{arrive}} = 0.03$. The team reward is defined the same in both modes, which is

Figure 9: Traffic junction environment.

$r_{\text{team}}^t = \sum_{i=1}^{n} r_i^t$, where $r_i^t$ is individual reward for car $i$ and defined as $r_i^t = C_i^t r_{\text{collision}} + \tau_i r_{time}$. Car $i$ has $C_i^t$ collisions (when two cars are in the same grid) at timestep $t$ and has spent $\tau_i$ in the junction. In addition, $r_{\text{collision}} = -10$ and $r_{\text{time}} = -0.01$. Each car has two actions, gas or brake, and follows specific route as illustrated in Figure 9 (*right*).

# B   Implementation Details

In cooperative navigation and predator prey, our model is trained based on MADDPG. The centralized critic and policy network are realized by three fully connected layers. The prior network has two fully connected layers. As for message encoder, two LSTM layers are used to encode messages. Leaky rectified linear units are used as the nonlinearity. The size of hidden layers is 128. For TarMAC and IC3Net, we use their default settings of basic hyperparameters and networks.

In traffic junction, our model is built on TarMAC. All networks (except for centralized critic, state encoder and prior network) use one fully connected layer. The centralized critic and prior network are realized by two fully connected layers. In addition, the state encoder uses gated recurrent units to encode trajectories, including historical messages and observations. Tanh units are used as the nonlinearity. The size of hidden layers is 128.

For threshold $\delta$ selection, we sort the causal effect $\mathcal{I}$ of all the samples in the training set and choose values of some certain percentile, *e.g.*, $90\%$, $80\%$ and $70\%$, as candidates, then we use grid search to get optimal $\delta$ among these candidates.

Table 4 and 5 summarize the hyperparameters used by I2C and the baselines in the experiments.

Table 4: Hyperparameters for cooperative navigation and predator prey

| Hyperparameter | I2C | MADDPG | TarMAC | IC3Net |
|---|---|---|---|---|
| discount ($\gamma$) | | 0.95 | | |
| batch size | | 800 | | |
| buffer capacity | $1 \times 10^6$ | | - | - |
| learning rate | $1 \times 10^{-2}$ | | $5 \times 10^{-4}$ | $1 \times 10^{-3}$ |
| $\lambda$ | 10 | - | - | - |
| $\delta$ (percentile) | $70\%, 80\%$ | - | - | - |
| $\eta$ | $1 \times 10^{-2}$ | - | - | - |

Table 5: Hyperparameters for traffic junction (medium and hard)

| Hyperparameter | I2C+TARMAC | TARMAC | IC3NET |
|---|---|---|---|
| discount ($\gamma$) | 0.95,0.90 | | |
| batch size | 40,80 | | |
| learning rate | $7 \times 10^{-4}$ | | $1 \times 10^{-3}$ |
| $\lambda$ | 10 | - | - |
| $\delta$(percentile) | 95% | - | - |
| $\eta$ | $1 \times 10^{-2}$ | - | - |

## C   Additional Experiment

We compare four different thresholds $\delta$ (90%, 80%,70%,50%) in prior network and evaluate how it affects the performance of model in the cooperative navigation. Figure 10 shows the learning curves in terms of final rewards of different $\delta$. As we can find out, I2C converges faster and is able to achieve higher reward as $\delta$ drops from 90% to 70%. Then I2C gets worse convergence as $\delta$ continues dropping. It is noted that the average communication overhead increases as value of $\delta$ decreases. Therefore, it turns out that too much communication contains redundancy and it can impair the convergence of model, and too little communication provides less valuable information for agents to make good decision.

Figure 10: Reward of I2C of different $\delta$ during training in cooperative navigation.