[Reviews · NeurIPS 2020]

Review 1

Summary and Contributions: This paper introduces I2C, a multi-agent communication architecture for cooperative tasks wherein each agent decides who to receive messages from. This is unlike prior work in multi-agent communication that has primarily focused on broadcast-style communication -- one/all agents sending messages to all other agents. The motivation is to reduce redundant communication (which might ease learning) and make the overall setup more practically realizable. I2C consists of a "prior network", which takes as input agent i's observation and predicts a probability distribution of which other agents to receive messages from. This prior network is trained with supervised learning to minimize the KL divergence between probability of the agent's action given the actions of agents other than i and probability of the agent's action given actions of agents other than i and j; the idea being that the prior network should enable communication only from those agents who might have a strong influence on agent i's action. This divergence measure is operationalized using the centralized critic as a function of joint action-value estimates. The rest of the network is similar to those in prior work (MADDPG, IC3Net, TarMAC) and consists of a message encoder, policy network, and a centralized critic. Experiments on cooperative navigation, predator-prey, and traffic junction gridworld settings demonstrate that I2C performs better than prior work and leads to sparser communication graphs than broadcast communicating MARL setups.

Strengths: This paper presents a novel, effective approach for learning who to communicate with for cooperative multi-agent tasks. This is an important problem -- broadcast-style communication can make learning harder (due to redundancy) and is hard to deploy in real applications. The authors present comparisons to a reasonable set of baselines and prior works -- full communication (with all observed agents), random communication, MADDPG, IC3Net, and TarMAC -- and show that not only does I2C get better task accuracies, it does so by using a reduced ratio of communication partners, which was the primary motivation for the proposed approach. The qualitative examples are interesting; especially for predator-prey, where agents show relatively sophisticated behavior of not "locking in" to the same prey and encirclement (when necessary).

Weaknesses: Across cooperative navigation and predator-prey experiments, it is unclear why IC3Net and TarMAC are worse than FC. FC allows communication between all agents within a visibility radius. TarMAC does the same, except that the receiving agent can use attention to decide which of the incoming messages are important vs. not. The IC3Net paper reports results on a predator-prey task as well on which it performs quite well. The primary difference, to my understanding, between the predator-prey from IC3Net and in this work, is that here there are multiple preys. Why does FC perform significantly better than IC3Net / TarMAC in this case? What is the performance of I2C on Traffic Junction? The result for I2C+TarMAC demonstrates that the key idea is complementary to prior works, but it would be useful to benchmark I2C alone as well, for a more direct comparison to TarMAC.

Correctness: Empirical evaluation largely seems sound. I've raised a few specific questions about this under Weaknesses / Suggestions that I'm looking forward to hear from the authors on.

Clarity: The paper is clearly written, figures are informative, and the proposed approach is well-motivated. Good work!

Relation to Prior Work: To my knowledge, key relevant prior works have been discussed and compared against.

Reproducibility: Yes

Additional Feedback: How well do agents perform on predator-prey if allowed to fully communicate with all other agents without any visibility radius? It might be useful to get a sense for what fraction of performance of this oracle approach can I2C get to. I understand that if requested, agent j will send a message to agent i, but it is unclear how "request" is exactly operationalized. Does request involve any scalar / vector sent from agent i to j? What is the architecture of the message encoder? How are the ID vectors d_i operationalized?


Review 2

Summary and Contributions: The paper presents a method for agents to selectively communicate messages to agents that are relevant and influential to the current agent’s decision-making. In order to decide which agents are influential, the agents maintain a prior that is learned via causal inference. The policy network is trained with a regularizer to ensure that the learned embedding for incoming messages approximates the effect of joint actions. UPDATE AFTER AUTHOR RESPONSE: I think the authors have adequately addressed the concern that IC3Net and TarMAC perform poorly in the cooperative navigation and predator-prey tasks. Citing Ref 24. in the author response and showing that the paper shares similar concerns about TarMAC was convincing. However, I would have appreciated more explanation as to why these architectures do not perform well with team rewards. I am happy to keep my score as is.

Strengths: + The ability for agents to selectively communicate by using causal inference is, to the best of my knowledge, a novel and useful contribution that can improve task performance. This seems sufficiently different from existing approaches like TarMAC. + The experiments are well-executed. The authors select helpful baselines that help the reader understand the effect of I2C. The authors also demonstrate that I2C can be used to reduce communication when combining with an existing algorithm TarMAC. +The paper is clearly written and the central contribution is easy to understand.

Weaknesses: -I am surprised that TarMAC and IC3Net perform so poorly, especially compared to MADDPG. More explanation on why this is the case would have been helpful. Could this have anything to do with suboptimal hyperparameter settings for these algorithms?

Correctness: Yes

Clarity: Yes

Relation to Prior Work: Yes

Reproducibility: Yes

Additional Feedback: Typo: line 312 “no much” -> “not much”


Review 3

Summary and Contributions: The paper proposes a novel method to learn agent-to-agent communication. Empirical results show that the method can not only reduce communication overhead but also improve agents’ performance.

Strengths: 1. The paper is well written in terms of clarity. Notations are well introduced and clearly defined. The empirical results are presented in a neat manner. 2. The interpretation of the learned policies and communication schemes, as well as the ablation study, makes the empirical results convincing. 3. How the method exploits the learned joint action-value function (sections 3.1 and 3.2) is novel to me, which might be interesting to the broader community in multiagent RL.

Weaknesses: Explanations for some key design choices in the empirical evaluation are missing. Please refer to my questions in “Additional feedback, ...”.

Correctness: My main concern regarding the correctness is the two-phase manner for training the prior network and its implications. Please refer to my questions in “Additional feedback, ...”.

Clarity: Yes, the paper is overall very well written in terms of clarity.

Relation to Prior Work: The authors discuss related work in learning communication in MARL, including DIAL, CommNet, IC3Net, TarMAC, etc.

Reproducibility: Yes

Additional Feedback: Questions: 1) As the authors point out, I2C is compatible with any framework of CTDE with a joint action-value function. Then, why did the authors build I2C on MADDPG for the first two domains, and another CTDE framework, TarMAC, for the third domain? 2) Presumably, the answer to my question 1) is that MADDPG is better than TarMAC in the first two domains. But why? Both MADDPG and TarMAC learn joint action-value functions, but since TarMAC allows communication it is supposed to be better than MADDPG. In line 234, the authors state that TarMAC is might be poor at handling tasks where team reward cannot be decomposed. Do the authors have additional evidence, possible from literature, to support this claim? Why MADDPG is good at it? 3) The authors report the empirical results where the prior work is trained in a two-phase manner. What does “two-phase” exactly mean? Is it that phase one is to train the prior network with data generated from an pre-trained CTDE algorithm, and phase two is to train the rest of the architecture with the prior network fixed? If that is the case, then it is not surprising that “the two-phase manner learns faster”, because it utilizes the pre-trained CTDE algorithm. 4) In line 201, the authors state that I2C and baselines are parameter-sharing. What does “parameter-sharing” exactly mean? What parameters are shared between I2C and, say, MADDPG? ----- I'm increasing my score from 6 to 7. Please incorporate the rebuttal into the new version.


Review 4

Summary and Contributions: The paper studies a multi-agent communication model where individual agents select each step which other agents they want to communicate with, and only communicate with those, sending a request and receiving an encoding of the other agent’s state in return. The setup follows the CTDE paradigm. Three tasks are used to obtain quantitative evaluations of the proposed model, all of which are fully cooperative, and the authors find that their method outperforms the baselines they compare to.

Strengths: The paper explores an interesting avenue within the multi-agent RL field, and the authors report promising results on benchmark tasks from the existing literature. The method is compared to and shown to outperform relevant baselines. Some ablations are included, with results indicating that the proposed method is indeed advantageous. The direction of research is motivated from both practical and academic perspectives: reducing communication overhead can be beneficial, and natural communicating groups (people, animals) operate in ways that become easier to model using a method like the one proposed in the paper.

Weaknesses: A discussion of the differences between artificial and natural systems with regard to communication is missing. It is clear that for humans communication overhead quickly becomes problematic, but agents are not like humans - they have much bigger potential communication bandwidth, and therefore probably have quite different optimal communication setups. Furthermore, the authors don’t investigate or motivate why their method outperforms alternatives where each agent communicates with all others. Given the scale of the tasks and teams of agents, does not seem obvious that excessive communication would impede task performance. COMMENT AFTER REBUTTAL: the authors' answer to the second point, that the agents have trouble identifying useful messags if they receive too many, sounds plausible. Inspection of the attention patterns could provide empirical evidence, which would be very helpful.

Correctness: Independent baselines are considered, standard tasks are used, and standard paradigms are used for parts that are not being innovated, making for fair comparisons. The versions of the tasks used are slightly different from the ones used in earlier works, making it difficult to precisely establish the correctness of the baseline scores being reported, but they look plausible.

Clarity: The paper is generally well-written, and the ideas and methods are clearly explained. In some places it could benefit from minor language editing (e.g. line 28 ‘form a good knowledge’, line 29 ‘many researches’, line 30 ‘the challenges aforementioned’), so I would recommend another thorough round of proofreading, but this does not significantly impede the quality of the paper in my view.

Relation to Prior Work: The authors cite relevant prior work and explain the point on which they differ. Explicit comparisons on performance are included with respect to MADDPG (Lowe et al, 2017), TarMAC (Das et al, 2018) and I3CNet (Singh et al, 2019), which themselves include comparisons to other earlier approaches such as DIAL and RIAL.

Reproducibility: Yes

Additional Feedback: I have several questions for the authors, which I think if addressed might flesh the paper out a bit. 1. Does it make sense to have multiple rounds of communication per timestep? 2. Wouldn’t it be helpful to allow for broadcasts as well, possibly with a penalty or budget for the number of allowed broadcast messages, that the agents have to learn to allocate? 3. What about sending unsolicited messages? The current proposal relies on agents being aware that other agents might have useful information for them; couldn't the converse also occur and be good to account for? 4. How different is it from TarMAC, given that there, agents effectively don’t attend to everybody else either? It seems to me that TarMAC should be able to reach the same solutions, do the authors have any insights with regards to explaining the observed differences?

[Author Response · NeurIPS 2020]

**Common Concerns.** For the performance of IC3Net and TarMAC on cooperative navigation and predator-prey, we were also surprised that they perform poorly, so we changed to optimizing individual rewards (easier settings) to check whether we operated these algorithms wrongly. The results showed they could converge and learned some elementary cooperative strategies. However, they cannot on team reward. Based on these observations, we hypothesize that they are hard to deal with team reward that cannot be decomposed. Both IC3Net and TarMAC are the official implementations. In addition, similar conclusions can be found in Ref.[24], "learning nearly decomposable value functions via communication minimization, ICLR 2020." They found that TarMAC struggles in all the scenarios of StarCraft II (cooperative game) and believed that this is because it cannot deal with the issue of reward assignment.

**Reviewer 1.** Thanks for your comments. As for difference in predator-prey between ours and the IC3Net paper, our setting has three moving preys, while IC3Net has only one prey and it is *stationary*, which means the task is extremely easy. Predators do not need to learn sophisticated strategy to encircle multiple moving preys. In traffic junction, our main purpose is to investigate the effectiveness of I2C on communication reduction, and thus we built I2C directly on TarMAC. We will benchmark I2C alone and add it up for a more thorough comparison in the final version.

If agents do not have any visibility radius, it means the environment is fully observable including all other agents. In this case, MADDPG can accomplish missions very well and it even converges faster than FC. Communication does not make much difference for no limited visibility because communication with other agents mainly gets their observations. And in fully observable environments, communication is not necessary since observation has already contained most of information from communication. In addition, too much redundant information could impair the learning.

For request-reply mechanism, it sends out a request (scalar) to the agent via a communication channel. Message encoder has two LSTM layers. The size of hidden layers is 128. In this paper, ID vector $d$ is set to its location info that an agent can access from its observation.

**Reviewer 2.** Thanks for your comments. We have tuned the key hyperparameters of I3CNet and TarMAC for better convergence, but it turns out their poor performance should have nothing to do with the hyperparameters. For the typos, we would correct them in the next version.

**Reviewer 3.** Thanks for your comments. In traffic junction, our main purpose is to investigate the effectiveness of I2C on communication reduction, and thus we built I2C directly on TarMAC, a communication method.

As for two-phase manner, phase one is to train the prior network with data generated from a pre-trained CTDE algorithm, and phase two is to train the rest of the architecture from scratch, with the prior network fixed. The two-phase manner learns slightly faster than end-to-end manner, but they converge to the same performance which is better than other methods. Another two-phase manner which shows similar results is that phase one is to train all the networks without prior network and use its action-value function to train prior network, and then phase two is to fine-tune the networks with the prior network fixed.

For the parameter sharing, we mean that all agents share their network weights.

**Reviewer 4.** Thanks for your comments. The literature has shown that excessive communication among agents could impede performance as in Ref [8, 21]. Our experiments also empirically verify this, comparing I2C with FC, IC3Net, and TarMAC. Our thought about the worse performance of excessive communication is that the agent can hardly learn which messages are useful, given excessive received messages. Even with attention like TarMAC, it is still hard.

Multiple rounds of communication per timestep may be required for explicit action coordination. However, as our work considers sharing encoding of observation between agents, it is not necessary to use multiple rounds of communication.

If "broadcast" means broadcast within the field of view, I2C can also achieve this if necessary (the agent can send the request to each of the agents in the field of view). If "broadcast" means broadcast to all other agents, this contradicts to our motivation since it leads to information redundancy that could even impair the performance, even with a budget of broadcast messages.

It is an excellent view for an agent to send the message to whoever it thinks the message is important for that agent. However, an agent cannot access the complete state information of other agents, therefore it is hard for the agent to infer whether it is really important for others. On the other hand, the agent knows exactly what situation it is in, so it is more reasonable to evaluate the others' influence on itself.

TarMAC receives messages no matter whether they are useful and uses attention to differentiate messages. I2C employs request-reply mechanism and learns to directly cut off messages they are not helpful. Intuitively, it is easy to learn whether a message is useful in terms of optimizing the RL target (I2C), while it is hard to weight all the received messages together in terms of optimizing the RL target (TarMAC).

We will have another thorough round of proofreading.

[Meta-Review · NeurIPS 2020]

All reviewers support acceptance of this paper, and I would also like to recommend acceptance. All reviewers point that this is an interesting a novel approach to learning who to communicate to in a multi-agent setup, which is both interesting from a research perspective but also useful in practical applications of multi-agent communication. Moreover, this paper is well executed, with clear statements supported by sufficient experiments and baselines. Finally, R1 and R2 have expressed concerns regarding the low performance of IC3Net and TarMAC. Authors have provided an explanation in the author response with some more experiments with regards to team vs individual rewards. I think these points should also be incorporated in the manuscript for completeness.